# Behavioural rhythms of two amphipod species *Marinogammarus marinus* and *Gammarus pulex* under increasing levels of light at night

Charlotte N. Underwood[1]◉*, Alex T. Ford[2,3]◉, Samuel C. Robson[3,4,5]◉,
Herman Wijnen[1]◉*

**1** School of Biological Sciences and Institute for Life Sciences, University of Southampton, Southampton, United Kingdom, **2** Institute of Marine Sciences Laboratories, University of Portsmouth, Portsmouth, United Kingdom, **3** School of Biological Sciences, University of Portsmouth, Portsmouth, United Kingdom, **4** School of Pharmacy and Biomedical Sciences, University of Portsmouth, Portsmouth, United Kingdom, **5** Institute of Life Sciences and Healthcare, University of Portsmouth, Portsmouth, United Kingdom

◉ These authors contributed equally to this work.
* c.underwood@soton.ac.uk; h.wijnen@soton.ac.uk

## Abstract

Artificial light at night (ALAN) is proliferating at an alarming rate across the globe, particularly around aquatic habitats. Natural and predictable light cycles dictate much of an individual organism's life by acting as a major signal for their circadian clock, driving rhythmic behaviours and physiological changes throughout the body. Light cycles also help populations coordinate group behaviour and greatly impact the inter-species dynamics of a community. Research into the ecological impacts of ALAN has highlighted numerous effects on these biological processes, including higher predation rates, impaired growth and development, and diminished reproductive success. Invertebrates play an undeniable role in ecosystem functioning and show robust daily rhythms. As such, it is vital to understand how ALAN may disrupt their behavioural patterns. The aim of this study was to monitor the impacts of increasing levels of light at night (0 lux – 80 lux), as well as constant light and constant darkness, on the behavioural rhythms of the intertidal amphipod, *Marinogammarus marinus*, and the freshwater species, *Gammarus pulex*. *Gammarus pulex* activity was not strongly synchronised to any of the light at night treatments. *Marinogammarus marinus*, however, exhibited strong behavioural rhythmicity in diurnal cycles with dark night periods. All the ALAN treatments resulted in a significant decrease in *M. marinus* rhythmicity and overall activity. Moreover, ALAN between 1–50 lux disrupted nocturnality in this species. These results indicate that while some amphipods show some adaptive plasticity when it comes to light pollution, others may experience strong direct impacts on their activity. This may be relevant to individual and population level fitness of vulnerable species in more heavily urbanised areas.

**Data availability statement:** All data are available from the University of Southampton Institutional Repository (doi: 10.5258/SOTON/D3024). The dataset relevant to this study can be found under the file name: Underwood_Thesis_Ch4data_Behavioural_rhythms_in_gammarids.csv.

**Funding:** This work was supported by Natural Environment Research Council (grant number NE/S007210/1). There was no additional external funding received for this study. The funders had no role in study design, data collection and analysis, decision to publish, or preparation of the manuscript.

**Competing interests:** The authors have declared that no competing interests exist.

## Introduction

The proliferation of the human population has led to a dramatic increase in artificial light at night (ALAN) around the world. More than 80% of people live under conditions that exceed natural night sky brightness [1–3] and the amount of artificially lit outdoor area is increasing by 2.2% each year [4]. The influence of artificial light extends beyond urbanised areas due to skyglow, the phenomena in which artificial light is scattered and reflected back to earth by the atmosphere. Skyglow is capable of masking lunar brightness [5] and has resulted in significant increases in light levels over 30 km from major cities [6]. As a result, more than 22% of coastlines are impacted by ALAN globally [7] and an estimated 35% of marine protected areas around the world experience ALAN [8]. Although there is currently no large-scale analysis of ALAN in freshwater environments [9], 90% of people live within 10 km of at least one substantial freshwater body such as a lake or river [10]. This means that ALAN has the potential to encroach on aquatic ecosystems both inland and towards the sea.

The widespread transition to broad-spectrum lighting like white LEDs will introduce ecosystems to night-time lighting at shorter wavelengths compared to older technologies [11]. This blue light is visually detectable by a wider range of taxa [11,12] and propagates more easily through the atmosphere and through water [13,14]. This increases the scope of light pollution in both terrestrial and aquatic habitats, an effect that carries many ecological implications.

The most thoroughly studied area of chronobiology has undoubtedly been the circadian clock, the biochemical oscillator set by the solar day. Consistent light cycles play a significant role in life's evolutionary history, shaping the daily rhythms found in nearly all organisms [15]. Light helps establish ecological niches, mediates physiological processes, and influences population dynamics and interspecies relationships [16]. Numerous studies have highlighted the ecological impacts of ALAN on behavioural and physiological processes known to be controlled by the circadian clock across a wide range of species, both terrestrial and aquatic [17]. Regardless of whether ALAN triggers a direct behavioural response in an animal, the presence of light at abnormal times can alter their natural activity patterns, and affect their ability to forage, find mates, or avoid predation [18–20]. While some species remain unaffected, many have been either positively and negatively impacted to some degree, with little understanding of the potential cascading effects and long-term consequences for ecosystems as a whole. For example, ALAN has been shown to modify communities through top-down and bottom-up effects [21,22]; to dissuade pollinators, thus limiting pollen transport in lit areas [23,24]; and to alter the composition of epifaunal marine communities [25]. By deterring certain organisms or providing more favourable conditions for some taxa over others, light pollution can upset the community structures and dynamics that have evolved over millennia [25,26]. This means that ALAN has consequences for individual fitness, as well as the intraspecies relationships, interspecies competition, and trophic interactions that make up an ecosystem.

Invertebrates are vulnerable to changes in light intensity and spectrum [27–30], and it is crucial to assess how their behaviour and physiology are impacted by ALAN and determine any long-term risks associated with chronic exposure. Amphipods are an order of crustaceans that have colonized virtually all aquatic ecosystems. The intertidal species *Marinogammarus marinus* (Leach, 1815; formerly *Echinogammarus marinus* [31]) and the freshwater species *Gammarus pulex* (Linnaeus, 1758), are commonly found throughout Europe [32,33]. Despite their significant ecological roles as nutrient recyclers and as a food source for higher trophic levels [34–42], there has been limited research into the impacts of light pollution on their behavioural and physiology. There has only been a single study highlighting its effects on drift rates in *G. pulex* [43]. Meanwhile *M. marinus* has not been included in any light pollution research to date. Yet there is evidence that light plays a significant role in modulating the physiology and behaviour of both species [44], especially in relation to other environmental stressors. Their typically negative relationships with light are known to be reversed by parasitism and exposure to certain pharmaceuticals [45–48], which leaves them at a potentially higher risk of predation in polluted waterways that are exposed to ALAN. There is also evidence that sex determination is linked to changes in photoperiod in *M. marinus* [49], meaning the proliferation of ALAN could have significant consequences for individual health, population dynamics, and distribution.

Based on these previous studies, it is important to understand how light pollution will alter the activity of *M. marinus* and *G. pulex* as it will directly interact with other behaviour-altering stressors, such as parasitism and chemical contamination. The aims of this research are to examine how increasing levels of ALAN affect the behavioural rhythms, activity levels and nocturnality in these two species of amphipod under increasing levels of light at night. The results from this work contribute to the development of a more holistic understanding of the compounding effects of environmental stressors impacting these ecologically important species.

## Methods

### Animal collection and husbandry

Adult *Marinogammarus marinus*, along with one of their main foods, the macroalgae *Fucus vesiculosus,* were collected by hand during low tide from Lock Lake, Portsmouth, United Kingdom (N 50°47'22.7", W 1°02'30.6") between July 2020 and February 2022. While *M. marinus* does often coexist with other amphipod species (e.g., *Marinogammarus obtusatus*; Dahl, 1938), they are the only species to ever be documented in Lock Lake [50,51]. *Gammarus pulex* were collected from River Ems, Westbourne, United Kingdom (N 50°51'34.8", W 0°55'45.8") using the kick sampling method [52] and 1 mm mesh nets between November 2020 and July 2021. On one occasion, *G. pulex* was collected from a culture at the University of Portsmouth, Institute of Marine Sciences, where the animals were contained in a 18L aquarium with water and organic debris collected from River Ems and then aerated with an aquarium pump and airstone. Half of the water in the aquarium was changed every three days and replaced with fresh river water. The aquarium was kept at 18°C under a 12h:12h light/dark cycle. Individuals from this aquarium were randomly chosen using an aquarium net. After collection, both *M. marinus* and *G. pulex* were transported to the University of Southampton, Life Sciences Building. No permits were required for this work as the sites are open to the public and neither study species falls under the requirements for ethics approval.

At the University of Southampton, each species was maintained in its own 5L capacity tank in an incubator set to 10°C and 70% humidity. For *M. marinus,* artificial seawater was prepared using Instant Ocean (Aquarium Systems, Sarrebourg, FR) and distilled water to attain a salinity of 33 ppt. *Fucus vesiculosus* was added to their tank to provide food and shelter. Water from River Ems was primarily used in the *G. pulex* tank, with Evian mineral water substituted in when river water was unavailable. Organic debris and sediment from River Ems were included in the tank to provide food and shelter. In both cases, the animals were allowed to feed ad libitum. The tanks were aerated using an aquarium pump and airstone, and then covered with a lid to limit evaporation. Half of the water in the tanks was changed twice per week and replaced

with new water. Wastewater was filtered through a 70 μm mesh to prevent animal loss. The incubator was equipped with a programmable light (Fluval Plant Spectrum LED) that allowed for gradual transitions from light to dark and vice versa over three hours. Amphipods were left in laboratory conditions for two weeks under a 12h:12h light/dark (LD) cycle prior to beginning any behavioural assays to allow for acclimation and for any tidal rhythms to dissipate in *M. marinus*. Daylight consisted of all LEDs on the Fluval light turned up to the maximum setting (2750 lux; 6500K).

**Behavioural assays**

To assess vertical movement in the water column, behavioural assays were conducted using Locomotor Activity Monitors (LAM25, TriKinetics Inc., Waltham, MA, USA), which count activity via the breaking of infrared beams (1 beam break = 1 count). Individuals were sexed based on their gnathopod size, which were enlarged on the males [53], and the uropod setae, which were hair-like on the males and spine-like on the females [54]. An approximately equal number of sexually mature males and non-gravid females (n = 32) were assigned to cylindrical vials (dim. 25 mm x 95 mm) filled with ~40 ml of either artificial seawater or river water, depending on the species of amphipod. The artificial seawater was first mixed with *M. marinus* tank water (2:1 ratio) to reduce its purity and minimise shock to the animals upon transfer to the vials. Prior to this, both seawater and river water were aerated with an airstone for 24 hours to ensure adequate oxygenation. During the assays, they were given a piece of food, *F. vesiculosus* for *M. marinus,* or organic debris for *G. pulex*. The vials were plugged with a cotton bung to minimise water loss from evaporation and were placed vertically in the LAM25. The activity monitors, along with a *Drosophila* Environment Monitor (DEnM, TriKinetics Inc., Waltham, MA, USA), were placed in an incubator set to 10°C and 70% humidity. All monitors were connected to a Dell Latitude D620 laptop located outside the incubator running DAMSystem3 *Data Collection Software* (TriKinetics Inc., Waltham, MA, USA) on Windows XP. The monitors were left for eight days to ensure a full seven days of uninterrupted data collection.

Both species were exposed to eight light regimes (Table 1). Natural night sky brightness ranges from 0.0001 lux under a cloudy night to 0.3 lux under a full moon [11,55]. The 12h:12h light/dark treatment (LD) was thus used as a control to simulate natural conditions. The first ALAN treatment (LA01) was selected to determine the lowest light level necessary to mask lunar light and elicit behavioural changes. 1 lux was the lowest light level the environmental monitor could detect. The remaining four ALAN treatments (LA05, LA30, LA50, LA80) were selected based on the range of light measurements taken from nearshore marine environments exposed to artificial light pollution [25,56,57]. Daylight was the same as the acclimation period (2750 lux; 6500K). A constant light (LL) and a constant dark (DD) treatment were also included to monitor free-running circadian behaviour for the first time in these two species (Table 1).

Cold white light (15000K) was used for the light at night treatments. We chose a white light treatment as one of the main concerns regarding the frequent use of white LEDs for street lighting around the world is that they will introduce

**Table 1. Light conditions for behavioural assays. Day phase length included the three-hour transitions at lights on/off, except under constant conditions (LL and DD).**

| Regime | Phase length (hours) | | Max. light intensity (lux) | |
| --- | --- | --- | --- | --- |
| | Day | Night | Day | Night |
| LD | 12 | 12 | 2750 | 0 |
| LA01 | 12 | 12 | 2750 | 1 |
| LA05 | 12 | 12 | 2750 | 5 |
| LA30 | 12 | 12 | 2750 | 30 |
| LA50 | 12 | 12 | 2750 | 50 |
| LA80 | 12 | 12 | 2750 | 80 |
| LL | 24 | 0 | 2750 | 2750 |
| DD | 0 | 24 | 0 | 0 |

more blue light into nightscapes compared to older technologies [3,11,58]. These short wavelengths have a comparatively stronger impact on behaviour and physiology [59–62]. Czarnecka *et al*. [63] found that freshwater amphipods from both light-naïve and urban habitats were both repelled by blue light.

### Activity analysis

After the eight-day period, the data was collected via DAMFileScan software (TriKinetics Inc., Waltham, MA, USA). The activity data was grouped into five-minute bins and then analysed and plotted using ClockLab Analysis 6 (Actimetrics, Wilmette, IL, USA). The data was normalised by dividing the individual results by the group mean and then averaging. This data was used to measure the rhythmicity, period length, total activity, and nocturnality of the amphipods. Rhythmicity was analysed using $X^2$-periodograms from the circadian range (15–35 h) with 0.01 significance levels. To determine if the individuals showed rhythmic behaviour, the relative rhythmic power (RRP) was calculated by taking the ratio of the peak amplitude divided by the significance threshold [64]. Behaviour was considered arrhythmic if the RRP was less than 1 (<1); weakly rhythmic if it was greater than or equal to 1 but less than 1.5 (≥1–1.49); and strongly rhythmic if it was greater than or equal to 1.5 (≥1.5). Only the period length of rhythmic animals was considered. Nocturnality was calculated as the proportion of activity counts during the designated night phase (ZT12:00 – ZT23:59), or the subjective night phase if conditions were constant (i.e., LL, DD; Table 1), divided by the total activity counts.

### Statistical analysis

Only data from individuals that survived the full eight days and were sufficiently active (total counts ≥ 200) was used in the statistical analyses. The data from one channel in the constant light treatment was also discarded as it was discovered at the end of the assay that the vial contained two *G. pulex* individuals. Due to high mortality of *M. marinus* under constant darkness, an additional assay was conducted to verify the results and enhance the statistical power of the analysis. *Marinogammarus marinus* and *G. pulex* data was analysed separately.

The percentage of animals that survived the assays, the percentage of surviving individuals that remained active, and the ratios of strongly rhythmic, weakly rhythmic, and arrhythmic individuals were each compared using Fisher's exact test. Normality of the behavioural parameters (total activity, RRP, period length of rhythmic individuals, and nocturnality) were tested using the Shapiro-Wilk test. The impacts of the night-time light treatment on those parameters were then analysed using non-parametric Kruskal-Wallis Rank tests followed by pairwise Mann-Whitney U-tests with Benjamini-Hochberg false-discovery rate corrections to adjust for multiple comparisons. For each light regime, the survival and proportion of active males and females were compared using Fisher's exact test. Mann-Whitney U-tests were also used to assess differences in total activity, RRP, period length, and nocturnality between the sexes. All statistical analyses were conducted using the software R (V4.0.2, R Core Team, 2020).

## Results

### Differences in female and male responses to light at night

Initial separate analyses for male and female individuals (S1 Table) uncovered little behavioural difference between male and female animals. This was the basis for pooling male and female data in the following analyses to obtain increased statistical power. The few significant sex-specific differences detected are outlined here for transparency.

*Marinogammarus marinus* males had lower survival and activity rates under LA80. Alternatively, while females had 100% survival under constant light (LL), the proportion of active females was only 50%. Their total counts were also significantly lower (Fig C in S1 Fig), and they became arrhythmic in contrast to the males, which were strongly rhythmic (S1 Table; Fig A in S1 Fig; activity profiles and actograms in S2 Fig). A Spearman's Rank Correlation test revealed a strong positive relationship between total counts and RRP under LL (rho = 0.676; p < 0.001), meaning the primary sex difference for *M. marinus* in LL might be that of reduced activity in females.

The proportion of active female *Gammarus pulex* was less than the males under LD and LA30. Females were significantly less active under LA01 and constant darkness (DD; Fig D-E in S1 Fig). They were also arrhythmic under LA30, whereas the males remained weakly rhythmic (S1 Table; Fig B in S1 Fig; activity profiles and actograms in S2 Fig).

## Percentage of living and active individuals

100% of *M. marinus* remained alive and active under the LD treatment. Comparatively, survival was significantly lower under four of the treatments (LA05, LA50, LA80, DD), with constant darkness yielding the highest mortality (Table 2). The proportion of surviving individuals that remained active was only significantly reduced in constant light (LL).

Overall, the average survival was higher for *G. pulex* (91.4%) compared to *M. marinus* (81.7%), but *M. marinus* had a higher percentage of active individuals (Table 2). The percentages of surviving *G. pulex* was not significantly affected by the light treatments, but the proportion of active animals was significantly higher under DD.

## Activity rhythms

Under a 12h:12h LD regime, *M. marinus* displayed activity that was strongly rhythmic (RRP = 2.37 ± 0.15; Table 2), with an approximate 24-hour cycle (period = 23.8 ± 0.17 hours; Table 2) and two activity peaks, one around lights on at ZT0 and a larger one right after lights off at ZT12 (Fig 1A–B). These peaks were either lost or dampened under the other light treatments, although they generally still occurred around the on/off transitions (activity profiles and actograms for all treatments in S3 Fig). Total activity significantly decreased (p < 0.05; Fig 2A) by an average of 55.4% under the ALAN treatments. Rhythms were significantly weakened (p < 0.05; Fig 2C) under all other conditions relative to LD, but only

**Table 2. Behavioural parameters of *Marinogammarus marinus* and *Gammarus pulex* activity under varying light/dark cycles.**

| Species | Light regime | $n_{tot}$ | $n_{al}$ | $n_{act}$ | $\%_{al}$ | $\%_{act}$ | Total daily counts | | Rhythmicity | | Nocturnality | | %SR | %WR | %AR | Period (hours) | |
|---|---|---|---|---|---|---|---|---|---|---|---|---|---|---|---|---|---|
| | | | | | | | Avg | ± SE | Avg | ± SE | Avg | ± SE | | | | Avg | ± SE |
| *M. marinus* | LD | 32 | 32 | 32 | 100.0[a] | 100.0[a] | 2613[a] | 242 | 2.37[a] | 0.15 | 0.74[a] | 0.02 | 90.6 | 9.4 | 0.0[a] | 23.8[a] | 0.17 |
| | LA01 | 32 | 30 | 29 | 93.8[b] | 96.7[ab] | 1598[b] | 245 | 1.50[b] | 0.16 | 0.44[b] | 0.03 | 34.5 | 44.8 | 20.7[b] | 24.2[ab] | 0.35 |
| | LA05 | 32 | 21 | 20 | 65.6[c] | 95.2[ab] | 864[c] | 297 | 1.35[b] | 0.12 | 0.49[b] | 0.03 | 30.0 | 40.0 | 30.0[bc] | 23.8[ab] | 0.60 |
| | LA30 | 32 | 30 | 29 | 93.8[ab] | 93.3[ab] | 1538[c] | 652 | 1.36[b] | 0.11 | 0.44[b] | 0.02 | 25.0 | 53.6 | 21.4[b] | 23.9[ab] | 0.18 |
| | LA50 | 32 | 26 | 24 | 81.3[bc] | 92.3[ab] | 1040[c] | 293 | 0.93[c] | 0.09 | 0.44[b] | 0.03 | 8.3 | 25.0 | 66.7[d] | 25.3[bc] | 1.04 |
| | LA80 | 32 | 23 | 22 | 71.9[c] | 95.7[ab] | 785[c] | 117 | 1.58[b] | 0.11 | 0.65[a] | 0.04 | 50.0 | 45.5 | 4.5[e] | 24.3[ab] | 0.55 |
| | LL | 32 | 31 | 22 | 96.9[ab] | 71.0[c] | 1534[bc] | 320 | 1.45[b] | 0.16 | 0.66[a] | 0.03 | 31.8 | 31.8 | 36.4[c] | 24.6[ab] | 0.95 |
| | DD | 64 | 34 | 28 | 53.1[c] | 82.4[bc] | 1052[c] | 377 | 0.93[c] | 0.04 | 0.48[b] | 0.03 | 0.0 | 32.1 | 67.9[d] | 26.6[b] | 0.83 |
| *G. pulex* | LD | 32 | 30 | 22 | 93.8[ab] | 73.3[a] | 1261[ab] | 189 | 1.26[a] | 0.12 | 0.55[ab] | 0.04 | 22.7 | 54.6 | 22.7[a] | 25.3[a] | 1.07 |
| | LA01 | 32 | 30 | 27 | 93.8[ab] | 90.0[abc] | 1972[b] | 321 | 0.88[bc] | 0.05 | 0.49[ab] | 0.02 | 0.0 | 33.3 | 66.7[b] | 24.5[a] | 1.42 |
| | LA05 | 32 | 29 | 27 | 90.6[ab] | 93.1[abc] | 1975[bc] | 309 | 1.01[bc] | 0.07 | 0.49[ab] | 0.03 | 11.1 | 37.0 | 51.9[c] | 25.9[a] | 0.97 |
| | LA30 | 32 | 32 | 22 | 100.0[b] | 68.8[ad] | 1157[a] | 322 | 1.25[ab] | 0.12 | 0.46[ab] | 0.04 | 31.8 | 27.3 | 40.9[d] | 25.5[a] | 1.44 |
| | LA50 | 32 | 26 | 21 | 81.3[a] | 80.8[abcd] | 870[a] | 133 | 0.92[abc] | 0.06 | 0.49[ab] | 0.03 | 0.0 | 38.1 | 61.9[b] | 25.3[a] | 1.19 |
| | LA80 | 32 | 27 | 23 | 84.4[a] | 85.2[abcd] | 1173[ac] | 204 | 1.12[abc] | 0.10 | 0.39[a] | 0.03 | 13.1 | 39.1 | 47.8[c] | 24.5[a] | 0.31 |
| | LL | 31 | 28 | 21 | 87.5[ab] | 75.0[abd] | 1164[ab] | 200 | 0.88[bc] | 0.06 | 0.54[b] | 0.03 | 0.0 | 33.3 | 66.7[b] | 25.4[a] | 1.77 |
| | DD | 32 | 31 | 30 | 96.9[ab] | 96.8[c] | 2266[b] | 337 | 0.79[c] | 0.04 | 0.47[ab] | 0.01 | 0.0 | 16.7 | 83.3[e] | 24.9[a] | 1.17 |

$n_{tot}$ = total number of individuals at the beginning of the assays; $n_{al}$ = # of individuals alive at the end of the eight day assays; $\%_{al} = (n_{al}/n_{tot}) \cdot 100$; $n_{act}$ = # of active individuals with total counts ≥ 200; $\%_{act} = (n_{act}/n_{al}) \cdot 100$; Avg = average; SE = standard error; %SR = % active individuals showing strongly rhythmic behaviour (RRP ≥ 1.5); %WR = % active individuals showing weakly rhythmic behaviour (RRP = 1–1.49); %AR = % active individuals showing arrhythmic behaviour (RRP < 1). Different superscript letters denote significant differences (p < 0.05) between treatments from the Fisher's exact tests and the Mann-Whitney U-tests. *M. marinus* and *G. pulex* were analysed separately.

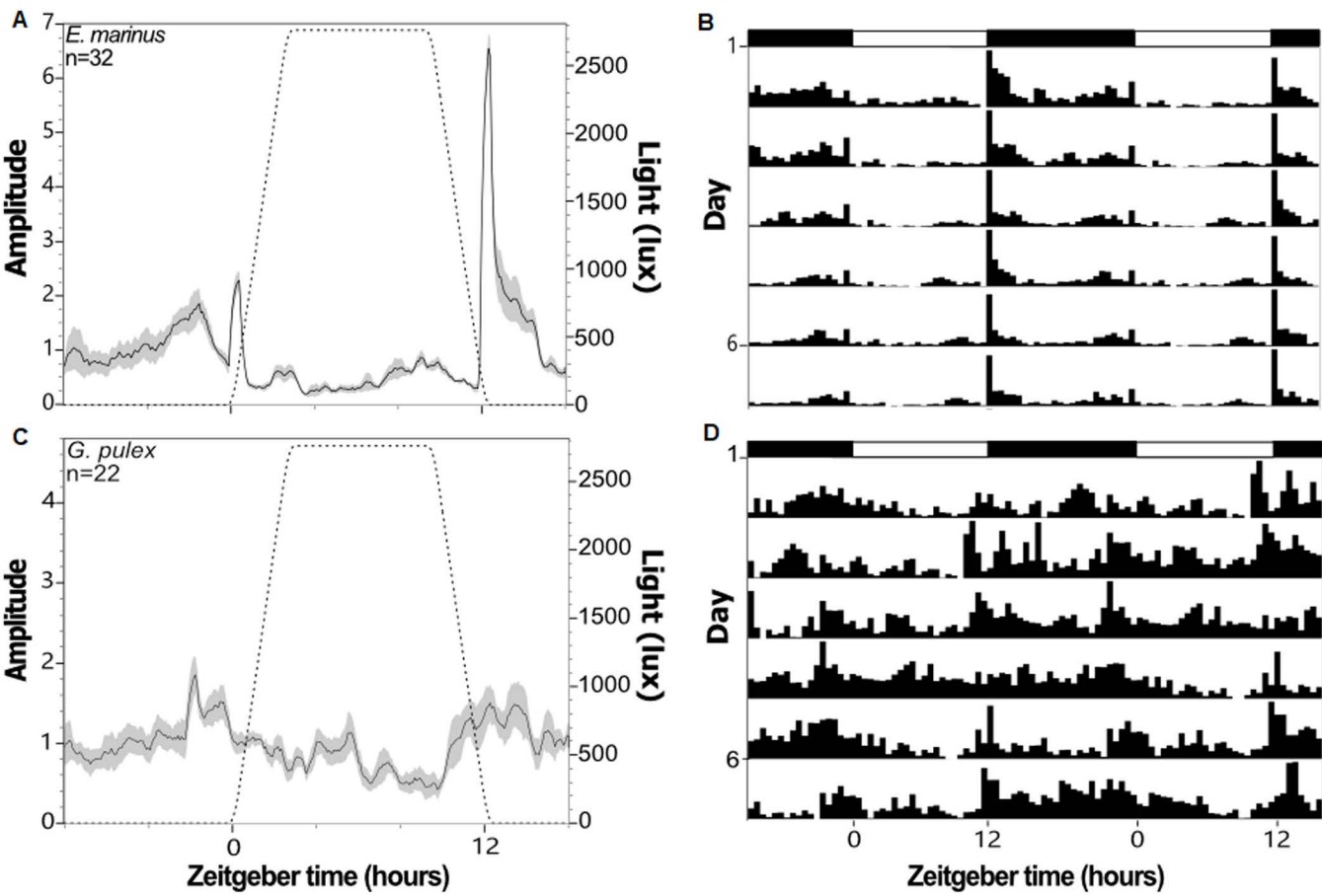

**Fig 1. Average activity profiles (left column) and double-plotted actograms (right column) of *Marinogammarus marinus* (A-B) and *Gammarus pulex* (C-D) under the LD light regime.** Activity profiles display average activity levels (black line) with standard deviation (grey areas), along with the light levels (dotted lines) over 24 hours. Actograms display the average, normalised behaviour across the 7-day assays, with each row showing two 24-hour cycles. Bars above actograms denote light levels; on/off transitions are not shown.

became arrhythmic (AR) under LA50. The ratio between strongly rhythmic (SR), weakly rhythmic (WR), and arrhythmic (AR) individuals significantly shifted under all light conditions compared to LD, moving towards more WR and AR individuals overall (Table 2). When rhythmicity was maintained, period length generally remained ~24 hours, although it should be noted that under LA50 and constant darkness (DD), when rhythmicity was relatively weak, a significant lengthening of the circadian period length was observed (Fig 2E). Under the LD regime *M. marinus* behaviour was predominantly nocturnal (0.74±0.02; Table 2), but they became more active during the day in the other light treatments, except LA80 and LL (Fig 2G).

Under a 12h:12h LD regime, *G. pulex* displayed activity that was weakly rhythmic (RRP = 1.26±0.12; Table 2) over an approximate 24-hour cycle (period = 25.3±1.07, Table 2), with two activity peaks, the highest around ZT22 and another at ZT12 (Fig 1C–D). These peaks were lost or dampened under all other light treatments, except LA80, although activity was generally higher around lights on (activity profiles and actograms for all treatments in S4 Fig). Total activity increased by 56.4% under the low ALAN treatments (LA01 and LA05) and by 79.7% under constant darkness (DD). Activity decreased at higher levels of light at night (LA30, LA50, LA80, and LL; Fig 2B). Behaviour became arrhythmic (RRP<1) under all but three light conditions (LA05, LA30, and LA80) but was only significantly lower under LA01, LA05, and the constant

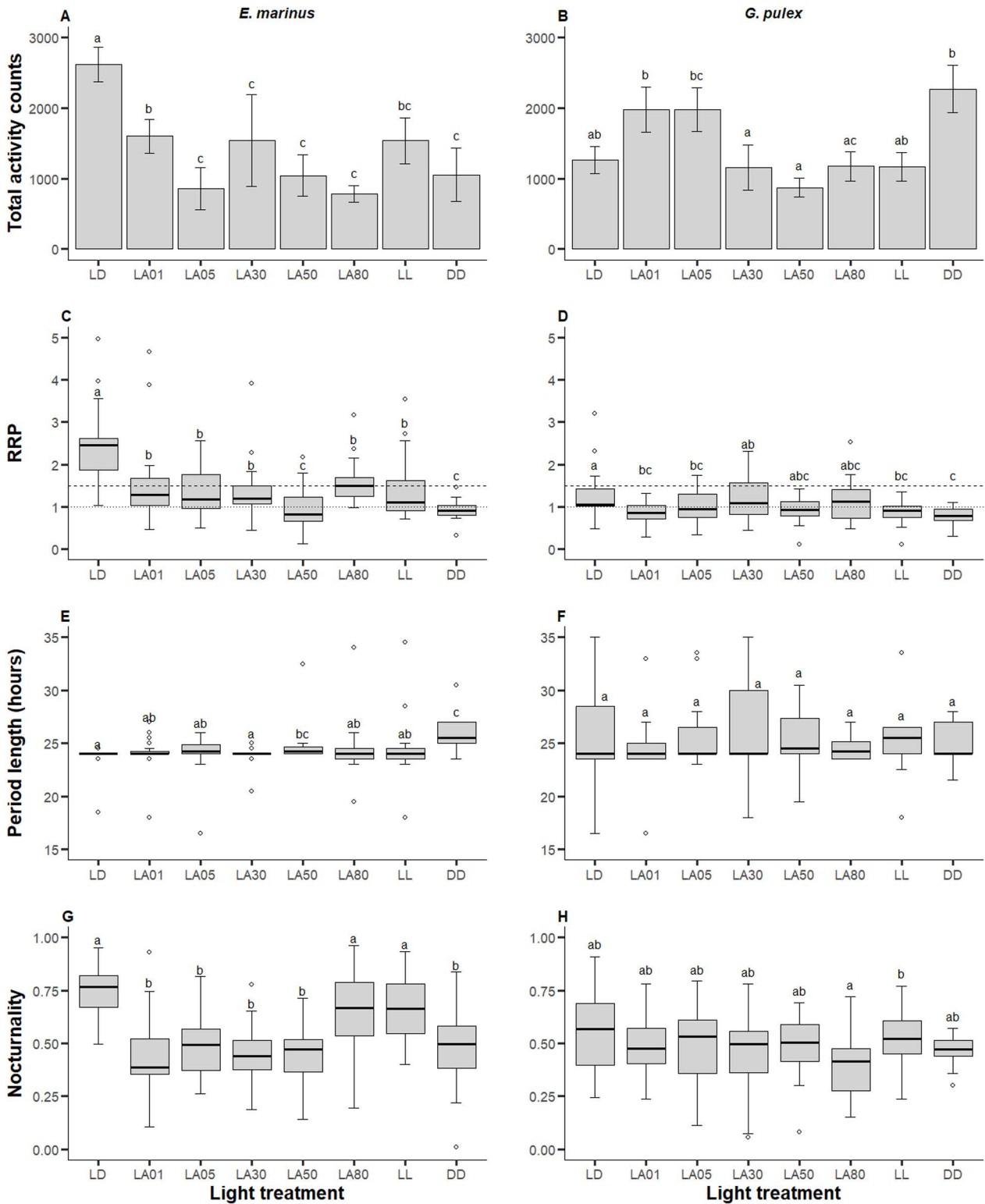

**Fig 2. Locomotor activity of *Marinogammarus marinus* (left column) and *Gammarus pulex* (right column).** (A-B) Barplots with standard error bars of total amount of activity counts logged during the seven-day assay period. (C-D) Boxplots of relative rhythmic power (RRP), which shows how strongly the activity repeated over a set period (e.g., 24 hours). An RRP ≥ 1 denotes rhythmic behaviour (dotted line); an RRP ≥ 1.5 denotes strongly rhythmic

behaviour (dashed line). (E-F) Boxplots of period length, which shows the length of time across which an activity pattern occurs in individuals with rhythmic behaviour (RRP ≥ 1). (G-H) Boxplots of nocturnality, which shows the proportion of activity that occurred during the subjective night phase (ZT12 – ZT23:59). Different lower-case letters indicate significant differences (p < 0.05) between treatments. Light conditions outlined in Table 1.

treatments (Fig 2D). The ratio between SR:WR:AR individuals, however, was significantly altered by the light treatments (Table 2). Although the period lengths showed high variance, it was not affected by the light treatments (Fig 2F). *Gammarus pulex* did not show a strong preference for either the day or the subjective night phase under any of the light conditions, although nocturnal behaviour was lowest under LA80 (Fig 2H).

## Discussion

The two amphipod species from this study demonstrated notably different patterns of activity under the LD regime, which was meant to simulate a natural 24-hour light/dark cycle. While both species were quite sensitive to even low light intensities, as illustrated by the peaks in activity around the light-dark transitions, *Marinogammarus marinus'* behaviour was strongly rhythmic over a 24-hour period compared to the weak rhythmicity of *Gammarus pulex*. Furthermore, *M. marinus* activity was clearly upregulated in dark phase, whereas *G. pulex* activity was more evenly distributed throughout the day. *Marinogammarus marinus* had a strong reaction to the different light treatments, lowering their activity, becoming less rhythmic, and in most cases, becoming more diurnal. Alternatively, *G. pulex* saw some increases in activity due to the light treatments, but rhythmicity remained weak or the behaviour became arrhythmic.

Under a few light conditions, both species showed reduced activity or rhythmicity in females compared to males. Notably, there were no significant sex-specific differences in the behaviour of *M. marinus* exposed to LD or ALAN cycles. Van den Berg *et al*. [65] determined that the interaction of size and sex could explain some, but not all, of the differences in swimming behaviour of *G. pulex* under light/dark conditions. On average, males are larger than females [53], and there is some evidence that females have shorter lifespans, likely due to the high cost of reproduction [66]. However, there is no conclusive evidence that these differences would elicit the contrasting behaviours between males and females witnessed in this study, particularly as they were not consistent across the light treatments.

While disparities between the two species in overall activity and their responses to light at night could be attributed to a number of factors, including size and ontogeny [65,67], we would argue that their differing foraging strategies and predator types are likely the main influences. *Marinogammarus marinus* live amongst one of their primary food sources, *Fucus vesiculosus* [68], and are predated upon by birds during low tide [39], and fish during high tide [69,70]. Minimising movement when exposed to light might be an adaptive response to avoid detection by predators. This would explain why there was a significant drop in activity under the ALAN treatments. It is unclear if a prolonged decrease in activity due to light pollution would affect the populations long-term fecundity as individuals spend less time looking for and engaging with mates. Likewise, the weakening of behavioural rhythms as a result of light at night may result in asynchronous activity amongst the population. Although this species experiences seasonal fluctuations in population [71], a sustained decline in numbers could lead to bottom-up effects for the community given their dual roles as detrivores and prey species [32,37,39,69]. Examining reproductive rates and sexual development under ALAN should be a priority for future research as it carries significant population and community-level consequences.

*Gammarus pulex,* on the other hand, does not live amongst their food source. They drift through their habitat in search of leaf litter [72], avoiding fish predation through continual movement [45]. Although presumed to be nocturnal [72,73], our results suggest that *G. pulex* does not necessarily favour the cover of darkness. Similar differences between the two species have previously been observed in regards to their phototactic behaviour, with *M. marinus* actively avoiding lit areas more than *G. pulex* [44]. Given that we found no differences between the ALAN treatments and the LD regime in terms of total activity, period length, or nocturnality, it is possible *G. pulex* could be more resilient to the impacts of light at

night. We cannot exclude that the increased levels of activity under LA01 and LA05 are reflective of increased foraging efficiency. Nevertheless, ALAN conditions might also increase vulnerability to visual predation [74]. Previous studies have found that light pollution has no prolonged impact on *G. pulex* drift rates [43], while exposure to predator cues does not elicit long-term reductions in activity and feeding rates [75,76]. Increased feeding observed in other freshwater amphipods exposed to light at night is hypothesised to be due to changes in metabolic rates as a result of light-induced stress [77,78]. This suggests that if *G. pulex* activity levels can recover after prolonged exposure to ALAN, it may actually be driven by a higher energetic demand. The prioritisation of foraging over predator avoidance regardless of environmental conditions has also been observed in dogwhelks, which were more likely to forage under ALAN regardless of the presence of a predator cue [79]. As with *M. marinus*, *G. pulex* are ecologically significant species, recycling nutrients back into the ecosystem and acting as a food source for higher trophic levels [34–36,42]. Further investigations into their activity, feeding, and metabolic rates under lower light levels are thus necessary to discern whether ALAN exposure is beneficial or detrimental to their individual fitness. This will help establish what cascading impacts this would have on freshwater habitats.

*Marinogammarus marinus* experienced significantly lower survivorship under three of the ALAN treatments. Exposure to ALAN can elicit increased rates of oxygen consumption [78], which would have depleted the limited oxygen in the vials. However, there was not consistently high mortality across all treatments with night-time light exposure. Under constant darkness, the unusually high mortality may be partially attributed to oxygen levels. Each vial contained a piece of *F. vesiculosus* to act as shelter and food for the animals throughout the assays. If the seaweed continued to photosynthesise under the other light conditions, this would have provided additional oxygen throughout the experiments, but this would not have been the case under constant darkness. Nevertheless, the low survival rates cannot be conclusively connected to the light conditions. Amphipods have previously been shown to be quite tolerant of low oxygen environments so the high mortality may have been unrelated [80–82]. Further investigations should include monitoring oxygen levels throughout the assays. Animals that did not survive or were insufficiently active were excluded from downstream analyses so the results presented here reflect the behaviour of live, motile animals.

Any level of light at night significantly weakened *M. marinus* behavioural rhythms. There was also a noticeable decrease in activity across all ALAN treatments. These results are consistent with previous work examining *M. marinus* escape responses to light [44], as well as the impacts of ALAN on talitrid amphipods whose activity was either arrhythmic [57] or reduced [19,56,83]. While ALAN did not lead to any changes in period length, *M. marinus* did appear to transition from predominantly nocturnal to more diurnal activity under the lower levels of light at night. Luarte et al. [19] saw a similar switch in the sandhopper *Orchestoidea tuberculata* (Nicolet, 1849) after seven days of observation, and shore crabs have been known to become less photonegative over time [84]. Due to oxygen limitations, we could not extend the length of the assays to determine if this switch became more pronounced over time, but these results suggest that some crustaceans may possess a certain level of resilience with regards to their relationship with light [85]. As with *G. pulex*, whether this versatility is truly an adaptive advantage or not remains to be seen. By becoming more diurnal under ALAN conditions, *M. marinus* may put themselves at risk of higher predation rates. There is evidence that amphipods can recover from short-term exposure to ALAN [57], but long-term light pollution would be a more relevant condition to study due to permanent coastal light sources.

These results also do not incorporate the potential influence of tidal cycles on *M. marinus* behaviour. Future studies may address how ALAN impacts amphipod behaviour during and after tidal entrainment. It is likely that exposure to ALAN would be gated by tidal cycles due to the light scattering and absorbing effects of sea water. Moreover, as described below, behavioural control has been found to exhibit altered light sensitivity when it is dominated by circatidal versus circadian timekeeping. Thus, ALAN impacts in the intertidal zone will likely be modulated from those in the absence of tidal entrainment. Intertidal organisms demonstrate clear rhythmicity in terms of their movement, feeding, and reproduction related to tidal regimes [86–88]. Several studies have highlighted that this circatidal clock operates independently of the circadian clock [89,90]. Following entrainment to tidal cycles, the hyalid amphipod *Parhyale hawaiensis* (Dana, 1853)

displayed peaks in activity correlated with high tide, regardless of daily light entrainment [91]. However, entrainment to daily light/dark cycles in the absence of tidal entrainment resulted in circadian rather than circatidal behaviour in this species. Under the latter circumstances. circadian rhythmicity was more pronounced in LL versus DD in *P. hawaiensis*, which matches our observations regarding *M. marinus*.

Other environmental stressors can influence amphipod behaviour as well. *Marinogammarus marinus* and *G. pulex* both act as intermediate hosts to behavioural-altering parasites, the prevalence of which can reach up to 70%, depending on location and time of year [45,46,92,93]. In addition, they inhabit areas which receive high levels of human pharmaceuticals via sewage effluent, with concentrations up to hundreds of micrograms per litre [94–96]. While these two species are typically photophobic, both parasitism and exposure to anti-depressant medications can reverse their relationship with light by modifying serotonergic activity in the brain [45,46,97]. The compounding impacts of these behavioural-altering stressors under increasing levels of light pollution could have serious impacts on amphipod survival as predation risk increases. This is of particular concern if the hunting efficiency of predators is enhanced by ALAN. As stated earlier, population declines could have cascading effects on the larger ecosystem.

While our results demonstrate significant behavioural changes due to light pollution, there is a high potential for ALAN to impact physiological processes as well. For example, photoperiod is a common factor in sex determination for invertebrates [98], including *M. marinus*, with longer days yielding higher rates of male broods during the summer months [49]. By obscuring the transition from day to night, anthropogenic light artificially extends the natural photoperiod and disrupts seasonal cycles in daylength. There is currently no information on how this could influence sex ratios at the population level over extended periods of time. Furthermore, disturbances to biological rhythms can lead to chronic stress [99,100], the physiological effects of which include altered metabolic function [78], tissue damage [100], and lowered immune function [101]. Measuring concentrations of hemocyanin, an oxygen-transport protein in the hemolymph of invertebrates [102], has successfully been used to monitor stress as a result of ALAN in talitrid amphipods [103] and may prove likewise suitable for gammarid amphipods.

Given their significant ecological role, it is important to understand the long-term impacts of ALAN on invertebrate behaviour and how it will interact with other environmental stressors, such as those considered above as well as the warming or acidification of the oceans due to climate change. Along with light, temperature is another important zeitgeber for the circadian clock [104–106] but, to our knowledge, there has been very little research investigating the potential compounding impacts of climate change and light pollution on biological clocks, including any behavioural or physiological effects [107–109]. Considering the confluence of anthropogenic stressors affecting the environment, more comprehensive studies examining the compounding impacts of said stressors should be considered. Our findings, along with studies that used even lower lux levels (<1 lux), assert that any amount of light at night that sufficiently masks the natural light-dark cycle can dramatically alter animal behaviour and physiology. This has resulted in less activity [110], altered foraging patterns [111], reduced melatonin production [112], and detrimental changes to reproductive success [18]. While higher levels of light pollution have also been linked to avoidant behaviour [113,114], increased rates of aggression [115], and significant shifts in predator-prey dynamics [74,116]. The breadth of behavioural and physiological consequences as a result of light pollution at any scale encourages more extensive research that focuses on how individual changes in fitness may be carried across generations, leading to long-term repercussions for population and ecosystem health.

## Conclusions

Light pollution is an ever-expanding environmental hazard in an increasingly urbanised world. This study illustrates that artificial light at night can have strong but disparate effects on the swimming response and behavioural rhythms across closely related species. We found significant decreases in activity levels, rhythmicity, and nocturnality as a result of ALAN in the intertidal *M. marinus* while the freshwater *G. pulex* remained largely unaffected. We also noted some potential sex-based differences that could warrant further investigation. Since these effects may be further exacerbated by other

anthropogenic stressors, field-based studies should be used to corroborate the results of our lab-based experiments in real-world conditions. By assessing a broad range of species under increasing levels of light at night, we can develop a robust methodology which can be used to help set regulations limiting the intensity or spectral range of artificial light, particularly around natural habitats. Given the evolutionary significance of natural light cycles in guiding a variety of biological processes, it is essential to understand how light pollution may shape coastal and riparian ecosystems in the future and mitigate those impacts through policy intervention.

## Supporting information

**S1 Table. Behavioural parameters of *M. marinus* and *G. pulex* activity under varying light/dark cycles.** Design is the same as in Table 2 of main manuscript but separated for female (f) and male (m) *M. marinus* and *G. pulex.*
(DOCX)

**S1 Fig. Activity of female and male *M. marinus* (left column) and *G. pulex* (right column).** Boxplots of relative rhythmic power (RRP; A-B) – how strongly the activity repeated over a set period (e.g., 24 hours). Dotted line = 1 (rhythmic); Dashed line = 1.5 (strongly rhythmic). Barplots of total counts (C-E) – total amount of activity counts logged during the seven-day assay period. Assays occurred over seven days with 3-hour ramping between light transitions. Asterisks denote significance levels from the Mann-Whitney U-tests (0.5*0.1**0.001***0.0001). Light conditions outlined in Table 1 of main manuscript.
(TIF)

**S2 Fig. Average activity profiles (columns 1 & 3) and double-plotted actograms (col. 2 & 4) of female and male *M. marinus* (col. 1 & 2) and *G. pulex* (col. 3 & 4).** Activity profiles display average activity levels (black line) with standard deviation (grey areas), along with the light levels (dotted lines) over 24 hours. Actograms display the average, normalised behaviour across the 7-day assays, with each row showing two 24-hour cycles. Bars above actograms denote light levels; on/off transitions are not shown. Y axes are the same within columns. Light conditions outlined in Table 1 of main manuscript.
(TIF)

**S3 Fig. Average activity profiles (columns 1 & 3) and double-plotted actograms (col. 2 & 4) of *M. marinus*.** Activity profiles display average activity levels (black line) with standard deviation (grey areas), along with the light levels (dotted lines) over 24 hours. Actograms display the average, normalised behaviour across the 7-day assays, with each row showing two 24-hour cycles. Bars above actograms denote light levels; on/off transitions are not shown. Y axes are the same within columns. Light conditions outlined in Table 1 of main manuscript.
(TIF)

**S4 Fig. Average activity profiles (columns 1 & 3) and double-plotted actograms (col. 2 & 4) of *G. pulex*.** Activity profiles display average activity levels (black line) with standard deviation (grey areas), along with the light levels (dotted lines) over 24 hours. Actograms display the average, normalised behaviour across the 7-day assays, with each row showing two 24-hour cycles. Bars above actograms denote light levels; on/off transitions are not shown. Y axes are the same within columns. Light conditions outlined in Table 1 of main manuscript.
(TIF)

## Acknowledgments

Thanks to the Invertebrate Facility of the School of Biological Sciences at the University of Southampton for providing space and equipment for the experiments that were conducted in this study.

## Author contributions

**Conceptualization:** Charlotte N. Underwood, Alex T. Ford, Samuel C. Robson, Herman Wijnen.

**Data curation:** Charlotte N. Underwood.

**Formal analysis:** Charlotte N. Underwood.

**Funding acquisition:** Alex T. Ford, Samuel C. Robson, Herman Wijnen.

**Investigation:** Charlotte N. Underwood.

**Methodology:** Charlotte N. Underwood, Alex T. Ford, Herman Wijnen.

**Resources:** Charlotte N. Underwood, Alex T. Ford, Herman Wijnen.

**Supervision:** Alex T. Ford, Samuel C. Robson, Herman Wijnen.

**Validation:** Charlotte N. Underwood.

**Visualization:** Charlotte N. Underwood.

**Writing – original draft:** Charlotte N. Underwood.

**Writing – review & editing:** Charlotte N. Underwood, Samuel C. Robson, Herman Wijnen.

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
