## [Decision Letter · Decision Letter 0]

Dear Dr. Underwood,

Thank you for submitting your manuscript to PLOS ONE. After careful consideration, we feel that it has merit but does not fully meet PLOS ONE’s publication criteria as it currently stands. Therefore, we invite you to submit a revised version of the manuscript that addresses the points raised during the review process.

We look forward to receiving your revised manuscript.

Kind regards,

Lubabalo Mofu

Academic Editor

PLOS ONE

Journal Requirements:

“Natural Environment Research Council (grant number NE/S007210/1) Research England’s Expanding Excellence in England (E3) Fund”

“This work was supported by the INSPIRE Doctoral Training Programme and the Natural Environment Research Council (grant number NE/S007210/1). Thanks to the Invertebrate Facility of the School of Biological Sciences at the University of Southampton for providing space and equipment for the experiments that were conducted in this study. SCR was partially funded by Research England’s Expanding Excellence in England (E3) Fund.”

“Natural Environment Research Council (grant number NE/S007210/1) Research England’s Expanding Excellence in England (E3) Fund”

5.  Thank you for stating in your Funding Statement:

“SCR was partially funded by Research England’s Expanding Excellence in England (E3) Fund.”

6. Please note that in order to use the direct billing option the corresponding author must be affiliated with the chosen institute. Please either amend your manuscript to change the affiliation or corresponding author, or email us at plosone@plos.org with a request to remove this option.

Reviewers' comments:

Reviewer's Responses to Questions

**Comments to the Author**

1. Is the manuscript technically sound, and do the data support the conclusions?

Reviewer #1: Yes

2. Has the statistical analysis been performed appropriately and rigorously?

Reviewer #1: No

3. Have the authors made all data underlying the findings in their manuscript fully available?

Reviewer #1: Yes

4. Is the manuscript presented in an intelligible fashion and written in standard English?

Reviewer #1: No

Reviewer #1: With respect

The study was re-examined statistically and content.

To further improve and improve the quality of study, do the following instructions.

Upon reviewing the manuscript titled "Behavioural rhythms of two gammarid species Echinogammarus marinus and Gammarus pulex under increasing levels of light at night", the following potential flaws were identified:

While the introduction provides a strong rationale, it lacks specific details about how the findings of this study advance the existing body of literature on artificial light at night (ALAN). Adding a clearer gap analysis could improve its positioning.

The introduction predominantly focuses on the negative effects of ALAN, with little acknowledgment of any adaptive or neutral responses by species to light pollution. This creates a slightly biased narrative.

The use of constant darkness (DD) as a control raises questions. The mortality observed in DD might have been influenced by oxygen deprivation, which was not controlled or monitored. Including oxygen level data for all conditions would strengthen the experimental rigor.

The study uses an equal number of males and females, but sex-specific behaviors were pooled for many analyses. This could obscure potentially important differences in how males and females respond to ALAN.

While the study mentions the impact of ALAN on tidal rhythms briefly, it does not experimentally test the interaction between tidal and circadian cycles. This limits the ecological applicability of the findings for intertidal species.

Statistical tests were performed appropriately, but the results could benefit from a more detailed explanation of effect sizes to demonstrate biological relevance beyond statistical significance.

The manuscript uses multiple post-hoc comparisons (e.g., Bonferroni correction). However, no justification for choosing this conservative method over others is provided. Some alternative corrections might allow more nuanced insights into the data.

Figures, while informative, are dense and require clearer labeling to aid comprehension. For example, Figures 2A-H include multiple parameters that could confuse readers unfamiliar with rhythmic activity metrics.

The results section references supplementary material (e.g., "Fig S1C"), which is crucial for interpreting key findings. These supplementary figures should be integrated more effectively or summarized in the main text.

While the discussion highlights key findings, it sometimes speculates without strong evidence (e.g., oxygen levels in vials, resilience of Gammarus pulex). These points should be either substantiated or rephrased to reflect their speculative nature.

There is limited focus on how findings can be applied to mitigate ALAN's ecological effects. Including suggestions for practical conservation measures or further research directions would enhance the impact.

No discussion is provided on how ALAN exposure in controlled laboratory settings reflects real-world ecological conditions. This is particularly relevant given the mention of skyglow and natural light scattering in the introduction.

The manuscript generally maintains a high standard of academic language. However, some sentences, particularly in the discussion, are overly long and complex, potentially hindering readability.

You can benefit from and cite the following studies to improve parts of the text:

- Knockdown resistance (kdr) associated organochlorine resistance in mosquito-borne diseases (Culex quinquefasciatus): Systematic study of reviews and meta-analysis

- Knockdown Resistance (kdr) Associated Organochlorine Resistance in Mosquito-Borne Diseases (Anopheles subpictus): Systematic Review Study

- Knockdown resistance (kdr) Associated organochlorine Resistance in mosquito-borne diseases (Culex pipiens): Systematic study of reviews and meta-analysis

Good luck.

**Do you want your identity to be public for this peer review?** For information about this choice, including consent withdrawal, please see our Privacy Policy

Reviewer #1: **Yes: ** Ebrahim Abbasi

---

## [Author Response · Author response to Decision Letter 1]

21 Mar 2025

We have attached a full Response to Reviewers as a separate file with our submission.

---

## [Decision Letter · Decision Letter 1]

Dear Dr. Underwood,

Thank you for submitting your manuscript to PLOS ONE. After careful consideration, we feel that it has merit but does not fully meet PLOS ONE’s publication criteria as it currently stands. Therefore, we invite you to submit a revised version of the manuscript that addresses the points raised during the review process.

Dear Dr. Underwood.

We look forward to receiving your revised manuscript.

Kind regards,

José A. Fernández Robledo, Ph.D.

Academic Editor

PLOS ONE

Journal Requirements:

Additional Editor Comments:

Dear Dr. Underwood.

The reviewer of the revised manuscript has some comments. Please address them and return the manuscript.

Sincerely,

-j

Reviewers' comments:

Reviewer's Responses to Questions

**Comments to the Author**

Reviewer #2: (No Response)

2. Is the manuscript technically sound, and do the data support the conclusions?

Reviewer #2: Yes

3. Has the statistical analysis been performed appropriately and rigorously?

Reviewer #2: Yes

4. Have the authors made all data underlying the findings in their manuscript fully available?

Reviewer #2: Yes

5. Is the manuscript presented in an intelligible fashion and written in standard English?

Reviewer #2: Yes

Reviewer #2: The present study deals with the alterations in the swimming behaviour of two gammarid species caused by ALAN. To address this issue, the authors acclimated and exposed the species Marinogammarus marinus and Gammarus locusta to increasing levels of artificial light, including light controls. The authors measured different behavioural parameters of both species, such as number of active individuals, period length of rhythmic or nocturnality, considering sex-specific responses. The topic of the study is manuscript is very interesting and necessary given the current illumination to which many shallow aquatic ecosystems are exposed. Furthermore, the manuscript is well-written, the experimental design is clear and robust, and the results are novelty, helping to fill a gap in the knowledge of the effects of light pollution on the organisms inhabiting the water column. Some important issues, mostly related to the experimental design, have been clarify in response to previous reviewers. There are, however, some points that must be addressed, for example, the quality of the introduction and also the method used by the authors to identify the species. I have also included some comments, suggestions and slight corrections below. For these reasons, I recommend the manuscript for publication pending these minor revisions.

Comments (all the comments below are also in an attached file)

Abstract line 37. The contracted form of a species cannot appear in the beginning of a sentence. Thus, “G. pulex” must be changed by “Gammarus pulex”. The same applies to “M. marinus” in the next sentence.

Abstract line 41. Replace “-“ with “–“ to use the same one than in line 35.

Introduction line 50. Replace “(4)” with “[4]”. In any case, I think that the paper that claimed the 2.2% increase is Kyba et al., 2017 (Artificially lit surface of Earth at night increasing in radiance and extent) instead of Davies et al., 2013.

Introduction line 53. I believe that this increase of illuminance must be considered as artificial rather than “natural light”.

Introduction line 75. Despite most of the mechanisms that cause the cascading effects are mostly unknown, I recommend going deeper into this point given the relevance in the structure of the community. Here I suggest some references dealing with these effects:

Bennie, J., Davies, T.W., Cruse, D., Inger, R., Gaston, K.J., 2015. Cascading effects of artificial light at night: resource-mediated control of herbivores in a grassland ecosystem. Philos. Trans. R. Soc., B 370 (1667), 20140131. https://doi.org/ 10.1098/rstb.2014.0131.

Bennie, J., Davies, T.W., Cruse, D., Inger, R., Gaston, K.J., 2018. Artificial light at night causes top-down and bottom-up trophic effects on invertebrate populations. J. Appl. Ecol. 55, 2698–2706. https://doi.org/10.1111/1365-2664.13240.

Sanders, D., Gaston, K.J., 2018. How ecological communities respond to artificial light at night. J. Exp. Zool. Part A: Ecological and Integrative Physiology 329 (8–9), 394–400. https://doi.org/10.1002/jez.215.

Introduction line 75: This sentence seems to reduce all the ALAN effects to phototaxis, but in the next sentence there are included some examples that are not necessary depending on movement alterations (e.g. disruption in mating could be caused by ALAN alterations in bioluminescence signals). Furthermore, movement alterations can also include disorientation, as Gaston and Bennie (2014) considered in their classification of the different organisms’ response to ALAN (i.e. light distracted species, light-exploiter species and light-repelled species). Therefore, I strongly recommend reformulating this statement to consider a wider spectrum of responses, or perhaps just replacing “regardless of whether an animal is positively or negative phototactic” to “regardless of whether an animal response to ALAN”.

Gaston, K.J. & Bennie, J. (2014) Demographic effects of artificial nighttime lighting on animal populations. Environmental Reviews, 22(4), 323–330. https://doi.org/10.1139/er-2014-0005.

Introduction line 78: In line 72 the authors mentioned the potential effects of ALAN in the community through cascading effects in some species. After that, the authors explained the potential alterations that ALAN can cause in the species, returning to the effects on the community structures and dynamics in the next sentence. I think that it would be clearer if the sentence “While some species remain unaffected…” were between “Regardless of whether an animal…” and “By providing more favourable conditions…”, but it is just a suggestion.

Introduction line 87. Synonymized names cannot be placed before authorship. Please, replace it with “Marinogammarus marinus (Leach, 1816) (formerly Echinogammarus marinus)”.

Introduction line 90. Is it possible to include some references to support their ecological role?

Introduction line 91. There is a study dealing with the swimming behaviour of Gammarus pulex. Although it does not include an ALAN-approach, they found different behavioural patterns between dark and light conditions, in case the authors would like to take it into consideration:

Van der Berg, S.J.P., Rodríguez-Sánchez, P., Zhao, J., Olusoiji, O.D., Peeters, E.T.H.M., Schuijt, L. (2021). Among-individual variation in the swimming behaviour of the amphipod Gammarus pulex under dark and light condition. Science of the Total Environment, 782, 162177. http://dx.doi.org/10.1016/j.scitotenv.2023.162177

Introduction line 95. Please, include some references to support the statement about the role of light in these species.

Introduction line 97. Change “…exposed to light at night” by “…exposed to ALAN”.

Introduction line 99. Change “…proliferation of artificial light” by “proliferation of ALAN.”

Introduction line 101. I think that this should be in plural since the authors have explained the potential effect of ALAN in these species using several papers.

Introduction line 108. After reading the whole Introduction, I think that this section would benefit from including additional examples related to the ecological impacts of ALAN, as I claimed in my comments above. There are several examples of specific and/or ecological alterations caused by ALAN in aquatic ecosystems that the authors could use to provide a greater context. Although it is true that the impact of ALAN in biological clocks still remains unexplored (which supports the relevance of the present study), a deeper understanding of how ALAN disrupts the ecosystem is mandatory for Introduction. I suggest this study by Sanders and Gaston (2018) but, as I said, there are examples of alterations in particular species, interactions or habitats that the authors could use.

Sanders, D., Gaston, K.J., 2018. How ecological communities respond to artificial light at night. J. Exp. Zool. Part A: Ecological and Integrative Physiology 329 (8–9), 394–400. https://doi.org/10.1002/jez.215

Methods line 111. This is the first time that the species in mentioned in this section, so it should be appeared as “Marinogammarus marinus”, but without the authorship.

Methods line 113. Replace it with “Gammarus pulex”.

Methods line 125. There are several species of this family in the British coasts (e.g. Marinogammarus obtusatus, Gammarus chevreuxi, Gammarus locusta, etc.), some of them coexisting with M. marinus. I believe that it is very important that the authors indicate in this subsection if they have used some key or the characters to identify these species and also if the differentiation from others in the same habitat was conducted in situ or in the laboratory.

Methods line 129. Replace it with “Fucus vesiculosus”.

Methods line 129. Since M. marinus is an intertidal species, were some pebbles or rocks from its habitat added to the bottom of the tank or just the algae? Although algae provide them shelter, the absence of their natural elements could stress them.

Methods line 140. Males and females are very similar in genera Marinogammarus and Gammarus with few exceptions that usually requires binocular stereoscope (e.g. setation in first antenna or shape of second gnathopod). Were both sexes differentiate based only on the body size or did the authors use sexual characters? Please clarify the method to differentiate sex.

Methods line 187. The authors use indistinguishably the terms gammarids and amphipods. Although in this context they are quite similar, they are not, so I recommend choosing one throughout the manuscript when referring to Marinogammarus and Gammarus.

Methods line 201. Was this high mortality caused directly by the effects of the darkness in the animal or indirectly by the death of algae?

Methods line 201. The authors explain that they discarded one channel since there were G. pulex in the vial of M. marinus. As I said above, I believe that it is necessary for the authors to include the characters to differentiate both species.

Methods line 203. Replace with “Marinogammarus marinus”.

Results line 223. Replace with “Marinogammarus marinus”.

Results line 225. In figure 1, 2 and S1 the species Marinogammarus marinus still appears as E. marinus. Please, replace them with “M. marinus”. Furthermore, in the caption of all the figures should appear the complete name of the species, not the contracted form.

Results line 231. Replace with “Gammarus pulex”.

Discussion line 313. Replace with “Marinogammarus marinus”.

Discussion line 314. Replace with “Gammarus pulex”.

Discussion line 316. Replace with “Marinogammarus marinus”.

Discussion line 322. In Van der Berg et al (2021) they discussed the observed differences in the swimming behavior of G. pulex according to sex with previous results in the literature. I believe that this point of the discussion would benefit from the proposed paper since there are few references to explain the lower activity in females.

Van der Berg, S.J.P., Rodríguez-Sánchez, P., Zhao, J., Olusoiji, O.D., Peeters, E.T.H.M., Schuijt, L. (2021). Among-individual variation in the swimming behaviour of the amphipod Gammarus pulex under dark and light condition. Science of the Total Environment, 782, 162177. http://dx.doi.org/10.1016/j.scitotenv.2023.162177

Discussion line 331. Replace with “Fucus vesiculosus”.

Discussion line 329. Since feeding and avoiding predation are probably the main contributions to the different light responses, there may be other factors that regulate their swimming behaviour and thus, the effect of ALAN. For example, Rygg (1972) studied the effect of light in the swimming behaviour of different gammarids from genera Marinogammarus and Gammarus and he observed differences during the ontogeny in some species. Therefore, if the population dynamic is not the same between both species this may cause slight differences. Of course, it is not possible to cover all the intraspecific variables in the experiment (indeed I consider robust the differentiation between males and females carried out in the present study), but I think that this paragraph should indicate that there may more factors behind the differences between both species.

Rygg, B. (1972). Factors controlling the habitat selection of Gammarus duebeni Lillj. (Crustacea, Amphipoda) in the Baltic. Annales Zoologici Fennici, 9(3), 172–183.

Discussion line 341. Replace with “Gammarus pulex”.

Discussion line 363. Replace with “Marinogammarus marinus”.

Discussion line 363. I agree with the authors, the low survival rate could be due to other factors unrelated to light. In fact, in some gammarids species there is cannibalism from the biggest males, even when algae are present. I do not know if the authors observed any sign of cannibalism in the dead specimens, but if so, it might be interesting to include this possibility in the potential causes.

Discussion line 378. Replace with “ALAN”.

Discussion line 385. Include the authorship: “Orchestoidea tuberculata Nicolet, 1849”.

Discussion line 405. Include the authorship: “Parhyale hawaiensis (Dana, 1853)”. Additionally, this species is not considered as talitrid but hyalid.

Discussion line 435. Discussion would benefit from including potential alterations in the community caused by the observed effects of ALAN in both species. Indeed, in Introduction section the authors rightly mentioned that light pollution has the potential to disrupt the communities through cascading effects on single species. However, little has been discussed about the ecological consequences of the suppression of swimming behaviour. Although this may be merely speculative, there is literature about the ecological role of amphipods (including gammarids) in benthic habitats, so I strongly recommend including a few lines of the potential consequences in the community or the ecosystem when they are affected.

Discussion line 443. Although this paper dealt with the combined effects of ALAN and ocean warming on productivity parameters (in copepods) rather than in circadian clocks, I think that the incorporation of this study may help with this part of the Discussion:

Nguyen, T.T., Le, M.-H., Doan, N.X., Pham, H.Q., Vu, M.T.T., Dinh, K.V., 2020. Artificial light pollution increases the sensitivity of tropical zooplankton to extreme warming. Environ. Technol. Innov. 20, 101179. https://doi.org/10.1016/j.eti.2020.101179

Conclusions line 455. This conclusion is focused on the generalize problematic of light pollution, rather than on the obtained results. Indeed, the only mention to the novel results is the following: “This study illustrates that artificial light at night can have strong but disparate effects on behavioural rhythms across closely related species”. The present study has revealed very interesting points about the swimming response of two similar species, including sexual differences, therefore, I do not understand why all these results are missing. Conclusion should be modified to include the novelties of the present work, keeping of course the issue of light pollution and the need to study a wider number of species in order to understand the true magnitude of this hazard.

**Do you want your identity to be public for this peer review?** For information about this choice, including consent withdrawal, please see our Privacy Policy

Reviewer #2: No

---

## [Author Response · Author response to Decision Letter 2]

9 Jul 2025

All comments have been addressed in the Response to Reviewers.

---

## [Editor Report · Decision Letter 2]

Behavioural rhythms of two amphipod species Marinogammarus marinus and Gammarus pulex under increasing levels of light at night

PONE-D-24-47613R2

Dear Dr. Underwood,

We’re pleased to inform you that your manuscript has been judged scientifically suitable for publication and will be formally accepted for publication once it meets all outstanding technical requirements.

Kind regards,

José A. Fernández Robledo, Ph.D.

Academic Editor

PLOS ONE

Additional Editor Comments (optional):

Reviewers' comments:

Dear Dr. Underwood

I did not handle this manuscript during the early submission process. I have decided to accept it based on R2 and after reviewing the responses to the reviewers' comments. I saw no need to resubmit it to the reviewers.

Sincerely.

-j